# Ultraprocessed Products as Food Fortification Alternatives: A Critical Appraisal from Latin America

**DOI:** 10.3390/nu14071413

**Published:** 2022-03-29

**Authors:** Maria F. Kroker-Lobos, Mónica Mazariegos, Mónica Guamuch, Manuel Ramirez-Zea

**Affiliations:** 1INCAP Research Center for the Prevention of Chronic Diseases (CIIPEC), Institute of Nutrition of Central America and Panama (INCAP), Guatemala City 01011, Guatemala; fkroker@incap.int (M.F.K.-L.); momazariegos@incap.int (M.M.); 2Nutrition and Micronutrients Department, Institute of Nutrition of Central America and Panama (INCAP), Guatemala City 01011, Guatemala; mguamuch@incap.int

**Keywords:** ultraprocessed products, obesity, fortification, food policy

## Abstract

Ultraprocessed products (UPPs), associated with obesity and non-communicable diseases (NCDs), are becoming predominant on the global market and a target for market-driven fortification initiatives. The aim of this article is to describe the implications of adding micronutrients to UPPs with excessive amounts of critical nutrients associated with NCDs and provide recommendations for legislation and policies. UPPs with added micronutrients such as breakfast cereals, sugar-sweetened beverages, powder beverages, fruit juices, sauces, and bouillon cubes, among others, are commonly available and heavily promoted in Latin American countries. Misleading advertising of UPPs with added micronutrients and with excessive content of sugar, fat, and salt might increase the consumption of such products, giving them a “health halo effect” that leads consumers to overestimate their nutritional quality and healthfulness. Although international collections of standards such as the *Codex Alimentarius* provide some guidelines on this matter, countries need to implement national legislations, through a food systems approach, to regulate the marketing and labeling of UPPs. Lastly, there is still the need to foster research to close knowledge gaps and help countries to guide the process of food fortification strategies from a regulatory standpoint.

## 1. Introduction

Low- and middle-income countries (LMIC) are facing micronutrient and protein/energy deficiencies but simultaneously are experiencing overweight or obesity in their populations, which are conditions that alone or in combination increase the risk of non-communicable diseases (NCDs) [1]. Changes in the global food system, triggered by the continues increment of food processed by industries, have exacerbated the increase in overweight globally [2]. In high-income countries (HIC), the consumption of ultra-processed products (UPPs) (e.g., foods attractive for their flavor and relative low price, but rich in salt, sugar, saturated fats, and refined cereals flours) provides more than 60% of the total daily energy intake [3,4], while in LMIC, per capita sales of these products have increased over 30% in the last 20 years [5,6,7].

Programs of large-scale food fortification (LSFF) (e.g., using commonly edible products manufactured by formal industries such as salt, oils, sugar, and cereal flours as vehicles to increase the supply of micronutrients) are widely recognized as a cost-effective public health intervention to reduce micronutrient deficiencies such as those of iodine, vitamins A and D, folate, and iron, among others [8,9]. Mandatory and voluntary programs of LSFF have been in place and continue being promoted in many countries [8,9,10]. Moreover, some UPPs, such as bouillon cubes and other condiments [11,12,13], are now being considered as potential vehicles for LSFF. Many of these industry-manufactured foods are also considered as UPPs with potential risks for human health [7,11,12]. On the other hand, market-driven fortification has been mentioned as one type of food fortification, although without specifying how to ensure its compatibility with public health [9]. A systematic review on food fortification examined 201 studies and found that, overall, processed products and UPPs such as noodles, infant complementary food, dairy products, and breakfast cereals were mostly used as fortification vehicles when infants where the target population [10]. Little is known, however, about the impact of the consumption of such fortified UPPs on the overall diet composition and NCDs-related outcomes in children and adults, especially when the vehicles contain excessive amounts of salt, fat, and sugar. The aim of this article was to describe the implications of using UPPs as acceptable candidates for micronutrient fortification and provide recommendations for legislation and policies on this subject.

## 2. Ultraprocessed Products and Risk of Non-Communicable Diseases

Using the NOVA system of food classification (Table 1), processed foods are essentially produced by adding salt, oil, sugar to other ingredients. Most processed foods contain two or three ingredients and are edible by themselves of in combination with other foods [13]. On the other hand, UPPs, such as pre-prepared dishes, ready-to-eat cereals (e.g., breakfast cereals), powder soups, powder beverages, bouillon cubes, and sauces like fish and soy sauces, are reconstituted products derived from foods and additives (preservatives, antioxidants, and stabilizers), with little if any intact ingredients [12]. The focus of this review is on micronutrients added to UPPs.

There is evidence linking the growing market of UPPs and NCDs. As the annual per capita volume sales of UPPs rise, the population adult body mass index increases [14]. Evidence shows that UPPs contain very high amounts of salt, fats, sugar, and simple starches, and hence they are highly energy-dense products in comparison with processed and minimally processed food. However, their content of fiber, good-quality protein, and plant-protective compounds is low [15,16,17]. Recent observational epidemiological studies, including systematic reviews and meta-analyses, have shown that the consumption of UPPs with excessive amounts of energy and critical nutrients (fats, saturated fats, trans fats, sodium, and free sugars) is associated with an increased risk of overweight, obesity, diabetes, cardiovascular diseases, cerebrovascular diseases, cancers, depression, and all-cause mortality in adults [18,19,20,21,22,23,24]. Results from a recent experimental study also revealed that the consumption of UPPs promoted an increment of calorie intake and weight gain [24]. There is increasing evidence that UPPs consumption is associated with diet-related NCDs, because most of these products have been formulated to induce excessive consumption by increasing their sensorial characteristics [14,25,26,27,28,29,30,31]. Evidence suggests that UPPs worsen the diets and health of children and adolescents by substituting breastfeeding, fruits, and vegetables and increasing the intake of salt, saturated fats, sugars, and simple starches, therefore having a negative impact on blood lipid levels and body fat [32,33,34,35,36]. A high consumption of ultraprocessed fruit juices, breakfast cereals, and processed milks has been shown to promote weight gain and the exceeding of the upper limits recommended for nutrients to limit or critical nutrients (free sugars, saturated fats, and salt) [25].

## 3. Health and Nutritional Claims on Ultraprocessed Products with Added Micronutrients

The food industry has introduced the addition of micronutrients to UPPs as a way to attract consumers to them [26]. This is uniquely problematic, as substantial evidence demonstrates the adverse health effects, including increased risk of diabetes, liver and kidney damage, heart disease, and some cancers, associated with the large consumption of these products [27,28,29,30,31]. Unfortunately, some UPPs (e.g., breakfast cereals, cereals bars, and juices) have become target products for micronutrient fortification [10,37,38,39], without considering the risk for NCDs linked to the increased consumption of these products. Table 2 shows examples of UPPs that are used or are being considered for micronutrient fortification commonly sold in Latin American countries, with excessive content of free sugar, total fat, saturated fat, and salt according to the PAHO nutrient profile [40,41].

Moreover, the food industry uses nutritional claims to add a positive message to some of the products mentioned above. Those claims include the nutritional content of a food (nutrition claims) or indicate that a relationship exists between a food and a health outcome (health claims) [42]. Health and nutritional claims are considered persuasive marketing techniques frequently present on food packages and used in all forms of advertising and promotion. Nutritional claims such as “Source of (Vitamin or Mineral)”, “High in (Vitamin or Mineral)”, among others, are frequently used by the food industry to influence purchase behaviors and food preferences [42]. Claims are likely to mislead consumers into believing that these products are healthier than those without claims, when in fact it may not be so, or they may even have a higher content of one or more critical nutrients whose intake should be reduced [42,43]. Convincing evidence has shown that the marketing of foods with undesirable nutrient composition increases the dietary intake and influences dietary preferences especially among children [43]. Children are the most vulnerable population to the effect of marketing as they lack sufficient nutritional knowledge and maturity to perceive the risks of their behaviors [44]. Moreover, the regulation of unethical and misleading advertising of UPPs marketing targeted to children has become a global target [45]. Here, it is important to point out that in Latin America and worldwide, UPPs are highly promoted, including through advertisements of their supposed nutritional and health-related properties, even when their nutrition quality is poor [16,46,47]. In Mexico, an analysis of 18,558 packaged products available in supermarkets showed that 39% of UPPs are being claimed as fortified with micronutrients. They include sugar-sweetened beverages (65%), ready-to-eat breakfast cereals (63%), and processed bread (65%). As expected, health and nutritional claims were found in higher proportions for products with micronutrient fortification in comparison with products without micronutrients (48.2% vs. 26.7%, *p* < 0.05, respectively). What is more worrisome, those fortified UPPs were more likely to contain excessive amounts of added sugar and artificial sweeteners [48]. Therefore, UPPs do not become healthy foods by the simply addition of micronutrients [49]. Until now, the lack of clear guidelines and regulations on food labelling in most countries is permissive of these abuses because it allows disregarding the overall nutritional quality of products [50].

## 4. Large-Scale Food Fortification and Market-Driven Fortification of Ultraprocessed Products

Large-scale food fortification (LSFF) uses basic food ingredients such as salt, oil, sugar, cereal flours, rice, and dairy products because they have a widespread consumption in target populations [51]. As these products are generic staple foods or condiments, they are not purchased by brand. In addition, one of the requirements of LSFF is that the intake of the fortification vehicles should not be increased, and therefore there is no need of advertising. However, “market-driven fortification” has been used by the food industry through a business-oriented initiative based on the incorporation of one or more micronutrients mostly to UPPs (e.g., ready-to-eat breakfast cereals, flavored milks and yogurts, cocoa powder, fruit juices, etc.) [9] and on their marketing as foods addressing micronutrient deficiencies and providing targeted health benefits [26]. The strategy of the food industries is to make health claims for their UPPs to move their products into the ‘health food’ business [49]. Market-driven fortification has become an effective marketing strategy as it is used to deflect attention from the quality of products and to promote the consumption of poor-quality UPPs [26]. The food industry has been aggressively marketing micronutrient-fortified UPPs in LMIC as a solution to micronutrient deficiencies [50]. For example, we found evidence in websites and supermarkets of fortified powder drinks, powder soups, breakfast cereals, and bouillon cubes (see Appendix A) in Central American countries [52,53,54,55,56].

The food industry have come under scrutiny in recent years over the production and promotion of UPPs and its contribution to poor-quality diets. These corporations have responded by reformulating products with less salt, fat, and sugar and increasing their micronutrient content. By doing this, they present themselves as part of the solution to the problems of both over- and undernutrition [26]. However, the great availability of processed products and UPPs with added micronutrients has risen some concerns, as there standards about which nutrients to add and in which amounts do not exist. First, there is a potential risk that unnecessarily high levels of micronutrients may be delivered to children if the same serving size of a fortified food (such as breakfast cereals, beverages, and nutrition bars) is intended for all members of a household [9]. Second, such UPPs could lead to an increased consumption of critical nutrients, without correcting the low intake of fiber and potassium [9].

Ready-to-eat breakfast cereals are one of the most commonly fortified UPPs marketed as containing a range of micronutrients [57]. Evidence from Latin America shows that market-driven fortification is used to market UPPs and that UPPs have in general poor nutritional quality. For example, a study in Guatemala found that 69% of ready-to-eat cereals displaying nutritional claims (including micronutrient-related claims) had a less healthy composition than products without such claims [58]. In Colombia, UPPs with a high content of salt, sugar, and trans-fatty acids also had a higher content of iron and folate than unprocessed and minimally processed foods [59]. The latter may be seen as a positive characteristic, but it does not correct the risk for NCDs associated with the overall composition of the products. Other analyses carried out in Colombia showed that 57.3% of analyzed ready-to-eat cereals had nutrient-content claims but at the same time were high in energy (93–99.9%) and sugar (88–93%) [60]. In Mexico, most of the UPPs with added micronutrients are baby foods (95%), ready-to-eat breakfast cereals (62.9%), and sandwich bread and tortillas (56.3%); these UPPs with added micronutrients were more likely to contain excessive free sugars and non-nutritive sweeteners [48].

## 5. Where Is the Balance for Public Health?

UPPs with added micronutrients may become contradictory with public health strategies aiming to improve diet quality and health outcomes. Some UPPs may be useful to reach vulnerable populations that do not have access to traditional LSFF; an example is provided by bouillon cubes, instead of salt, in many Sub-Saharan countries (Box 1) [61]. However, this practice must be accompanied by rules and regulations to prevent the increment of the consumption of these products and to avoid presenting them as “healthy” foods or introducing them in countries that do not require them.

A regulation of fortified UPPs, accompanied by population nutritional education, is thus necessary not only to ensure that the intake of micronutrients remains in useful and safe ranges but also to limit the intake of critical nutrients (e.g., salt, saturated fats, and sugar) [9]. Currently, nutritional policies are mostly intended to encourage product reformulation by reducing the use of critical nutrients. There is a need for stronger regulations that effectively guide claims and advertising messages, as well as specify the practices of adding micronutrients to these products. However, it is important to consider that reformulation by itself, and specifically, the addition of micronutrients, will not turn UPPs into healthy products [62]. Thus, strategies to regulate the addition of nutrients and control health claims and, most important, the marketing all UPPs need to be considered [50,62,63].

Box 1Fortification of condiments, seasoning, powder soups, and sauces.Condiments, seasonings, and powder soups are affordable and regularly consumed; therefore, they are food vehicles for micronutrient fortification. For example, many of Nestlé’s range of “Popularly Positioned Products”, which are targeted at low-income consumers, are fortified with iron, iodine, zinc, or vitamin A; an example is provided by the affordable and popular Maggi^®^ products (sauces and seasoning cubes) [53,54,64]. These products are major sources of sodium (e.g., fish and soy sauce, bouillon cubes, powder soups) or added sugars (e.g., ketchup, tomato paste) [40]. Although condiments, seasonings, and sauces might be appropriate fortification vehicles under certain circumstances and in certain countries (based on their patterns of consumption), their promotion (market-driven fortification) should be restricted, and future research is needed to assess the potential risks of these fortification initiatives [65].

### 5.1. Food Systems Approach to Regulate the Labelling and Marketing of Ultraproccessed Products with Added Micronutrients

In LSFF or market-driven fortification of UPPs, the use of nutritional and health-related claims must be regulated because specific references to nutritional and health claims associated with micronutrients in certain products might pose a threat to public health, as they increase the consumption of unhealthy foods, especially by children [42,43]. For example, UPPs feature misleading health and nutrition claims on their packages, e.g., related to a particular nutrient (e.g., “high in calcium”) and giving a “health halo effect” that leads consumers to misunderstand and overestimate their nutritional quality and healthfulness [66,67,68]. A food systems approach, through a front-of-package (FOP) labelling and marketing regulation, is needed to ensure the protection of public health and the consumer’s right to information. For example, some Latin American countries, such as Chile, Peru, and Mexico, have implemented the FOP warning label system, using a black warning octagon when a product is “high in” sugar, salt, saturated fat, total fat, and trans fat. In Chile, products with such warnings cannot display any nutritional or health claim, including references to specific micronutrients [69]. Policies monitoring and restricting nutrition and health claims on UPPs high in salt, fat, sugar, and energy will improve food [70]. Fortunately, the number of transnational food and beverage corporations that are flooding the markets of LMIC with UPPs is small. Since most national food industries (small and medium) generate less processed food, they could become allies in transforming food systems into sources of healthy and accessible options for all ages [63]. Hence, the nutritional quality of the food supply might be improved by incentivizing and reformulating community and local food production [70].

### 5.2. Need to Implement National or Regional Regulations

The Principles for the Addition of Essential Nutrients to Foods by the *Codex Alimentarius* provide guidance for the safe addition of nutrients to foods to be used by countries in the development of their own policies and regulations based on national or regional contexts. The development of these principles took into consideration the nutritional risk analysis approach, which includes the risk to health posed by an inadequate and/or excessive intake of nutrients or related substances and how a food matrix could increase the risk of an adverse health effect in the presence of a nutrient with potential beneficial effects [71,72]. In the context of UPPs and their fortification, these principles are useful when evaluating the risk of adverse effects derived from the high content of critical nutrients present in their food matrix, which give the consumer a misleading message about the a product’s health benefit. On the other hand, the Codex does not mention specific foods to which nutrients may not be added, but rather recommends that competent authorities in a country or at the regional level should determine those. Codex only states clearly that nutrients should not be added to alcoholic beverages, and this has been included in regulations in the European Union, United States, Australia, New Zealand, and Chile [73,74,75,76,77].

In general, policies, standards and/or regulations regarding food fortification in different countries aim to prevent nutritional deficiencies through the addition of micronutrients to foods. However, aspects such as the regulation of market-driven fortification to prevent the excessive intake of nutrients or UPPs that are not appropriate for fortification due to their high content of critical nutrients are seldom considered. The regulation of the addition of nutrients to certain foods is not new and has been addressed in several countries. In the United States, the fortification policy in the Code of Federal Regulations does not “consider appropriate to fortify snack foods such as candies or carbonated beverages and does not encourage the indiscriminate fortification of foods”. The policy does not include more requirements for the fortification of other UPPs and refers to particular regulations for specific foods or classes of food [77]. The Code of Food and Drug Regulation of Canada describes a list of foods that may be fortified and also specifies which nutrients can be added. Most of the foods are staples or processed foods, but the list also includes UPPs such as ready-to-eat breakfast cereals or other breakfast or meal replacements foods [78]. The Chilean regulation prohibits the fortification of a list of products that include confectionery products or similar, such as chocolates, candies, chewing gums, sweets, cookies, ice cream, jams, and carbonated beverages; however, it allows the fortification of non-carbonated beverages and powders to prepare beverages [76].

The policy guideline of Australia and New Zealand states that the “permission to fortify should not promote increased consumption of foods high in salt, sugar or fat or foods with little or no nutritional value that have no other demonstrated health benefit”, and the requirements for the application of the policy are described in their specific standards. Fortification of UPPs such as biscuits, ready-to-eat breakfast cereals, ice cream or analogues, among others, is allowed only with specific vitamins and minerals listed in the standard collection and if they comply with either the maximum or the minimum content established for specific nutrients like fat, sugar, or protein. In the case of vitamin D, breakfast cereals must meet the nutrient profiling score criterion to be fortified. The scoring method uses the content of energy, saturated fats, sugars, salt, fiber, and protein as well as the use of ingredients such as fruits, nuts, vegetables, and legumes in the formulation to calculate the score of a product [73,74]. Currently, the standard only specifies the application of the scoring method to breakfast cereals, probably because of they are broadly consumed as the main source of nutrients in the morning. This is a good example of how a nutrient profile score could be used to promote the reformulation of UPPs and allow for their fortification. However, it should be applied cautiously and not to all UPPs, but to those that are the main or important nutrient sources. More recently, an amendment to the regulation of food fortification in India states that foods with a high content of critical nutrients shall be excluded from fortification. However, the definition of high content of these nutrients has not been provided, and compliance is not in effect yet [79,80]. India has considered the experience of Chile, Mexico, and South Africa applying limits for sugars, fat, and salt using the FOP warning label system to include them in the fortification regulation [79].

In 2002, the Institute of Nutrition of Central America and Panama (INCAP) issued a proposal for regulating voluntary fortification, which included a list of foods that should not be fortified based on their nutritional profile [81]. However, the adoption of such regulation requires not only technical knowledge but also a political will at the country level and a leading role of the Ministries of Health in order to protect consumers’ health, which needs to be strengthened. The proposal was updated in 2015 and is currently under review. Advocacy to implement regulations regarding voluntary fortification that consider the aspects discussed in this paper must continue. The INCAP proposal may be used as a guideline for countries with similar contexts as Central America.

In summary, although the principles for addition of nutrients and the selection of foods have been presented in international standards such as Codex, countries must take actions through national legislations to develop and implement these regulations in a coherent way, with efforts to prevent UPPs consumption.

### 5.3. Fostering Research to Close Knowledge Gaps

The exploration of a causal association between consumption of fortified UPPs and concomitant rise in obesity, cancer, diabetes, and other NCDs is an interesting area for future research. Currently, there is very limited evidence of the impact of fortified UPPs on health outcomes, and no studies have addressed yet the overall composition of fortification vehicles (e.g., bouillon cubes, soy sauces, powder soups, etc.), generally comprising high amounts of salt and saturated fats, and their impact on NCDs in large-scale fortification programs aimed at reducing micronutrient deficiencies. Future knowledge on this matter could help countries to guide food fortification strategies from a regulatory standpoint.

## 6. Conclusions

The coexistence of undernutrition, micronutrient deficiencies, and excess weight and the growing consumption of UPPs imposes the need to rethink fortification initiatives, either LSFF or market-driven fortification. Governments need to ensure a fortification legislation that is complementary (not antagonist) to other public health policies such as salt and sugar reduction strategies. That is, the addition of micronutrients should not result in adverse effects for health such as obesity, diabetes, high-blood pressure, and other NCDs. Adding micronutrients to UPPs is in the center of this discussion, because of the recommendation originated from the overwhelming epidemiological evidence of the association between fortified UPPS and the risk of NCDs as well as the increase in all-cause mortality, especially in countries with high burden of such diseases. Although international standards collections such as Codex have provided some guidelines, each country needs to go beyond Codex through national legislations, regulating the types and content of micronutrients, ingredients, and nutrition- and health-related claims and restricting the marketing and availability of UPPs. There is an urgent need of strengthening the regulation capacity of the Ministries of Health for UPPs fortification legislation in LMIC countries with high prevalence of NCDs. Finally, some questions remain unanswered, calling for further research exploring the association between consumption of UPPs with added micronutrients and the concomitant rise in obesity, cancer, diabetes, and other NCDs, along with the evaluation of the impact of existing large-scale fortification programs using vehicles such as sugar and salt, which are ingredients that have been fortified with micronutrients that are deficient in the population, but whose consumption must not be promoted to prevent the excessive intake of critical nutrients.

## Figures and Tables

**Table 1 nutrients-14-01413-t001:** NOVA food system classification according to levels of processing.

Group	Definition	Examples
Unprocessed or minimally processed foods	Unprocessed foods are natural foods that are edible parts of plants or animals. Minimally processed foods are natural foods altered for preservation or storage. Also include meals cooked at home or in restaurant kitchens as dishes or meals.	Seeds, fruits, vegetables, leaves, stems, roots, meat, eggs, milk, frozen vegetables, legumes.
Culinary ingredients	Substances derived from minimally processed foods or from nature by processes to obtain durable products that are suitable for use in home and restaurant kitchens to prepare, season, and cook. They are not meant to be consumed by themselves.	Salt, sugar, oils, butter.
Processed foods	Contain two or three ingredients and are recognizable as modified versions of unprocessed or minimally processed foods. They are edible by themselves or, more usually, in combination with other foods.	Bottled vegetables, canned fish, fruits in syrup, cheeses, and freshly made bread.
Ultraprocessed products	Made from substances derived from foods and additives, with little if any intact natural ingredient. They have a long list of ingredients, and many are derived from further processing of food constituents, such as hydrogenated or inter-esterified oils, hydrolyzed proteins, soya protein isolate, invert sugar, and high-fructose corn syrup. Include additives to imitate or enhance the sensory qualities (e.g., sodium monoglutamate) to create more convenient (ready-to-eat) and attractive products.	Soft drinks, cereal bars, breakfast cereals, industrial bread, sweet or savory packaged snacks, reconstituted meat products, pre-prepared frozen dishes, fast food, powder mixes, ready-to-eat food.

**Table 2 nutrients-14-01413-t002:** Ultraprocessed products with added micronutrients and their nutritional profiles, commonly sold in Latin American countries.

Group	Product ^a^	Critical Nutrients in Excess ^b^
Ready-to-eat cereals	Cereal bars and breakfast bars	Free sugars, total fat, saturated fat, and salt
	Breakfast cereals	Free sugars and salt
	Nutritional bars or “energy” bars	Saturated fat
	Instant noodles	Total fat, saturated fat, salt, and free sugar
	Packaged bread (industrial)	Salt
Sugar-sweetened beverages	Sports beverages	Free sugar and salt
	Energy drinks	Free sugars
	Other juices and fruit drinks	Free sugars
Oils	Oils and spreadable fats	Total fat, saturated fat and salt
Soups	Dehydrated soups	Saturated fat and salt
	Instant soups	Saturated fat and salt
Dairy	Sugary Milk beverages	Free sugars and saturated fat
	Liquid flavored yogurt	Free sugars
	Semi-liquid flavored yogurt	Free sugar and saturated fat
Sauces, seasonings, and condiments	Liquid sauces	Total fat, saturated fat, and salt
	Soy Sauce	Total fat and salt
	Pasta sauces	Salt
	Ketchup	Free sugar and salt
	Bouillon cubes and bouillon powders	Total fat, saturated fat, and salt
	Other sauces and condiments	Free sugar and salt
Food for children <2 y	Products targeted to babies and infants	Free sugars, saturated fat, and salt

^a^ Products marketed with added micronutrients mostly by voluntary initiatives. ^b^ Excessive amounts of critical nutrients based on the Pan American Health Organization (PAHO) nutrient profile [41].

## Data Availability

Not applicable.

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
