# Peer review of "Ultraprocessed Products as Food Fortification Alternatives: A Critical Appraisal from Latin America"

_nutrients, 2022, doi:10.3390/nu14071413_

Round 1

Reviewer 1 Report

This is a generaly useful narrative review of a topic that is generating interest and concern.

The authors appear to have addressed this review with an assumption that UPP is a 'Bad Thing'.  It is not always.  A more balanced approach would be preferred.

A fuller explanation of what food processing includes is needed.  Some food processing is essential (eg washing, and peeling, gutting and butchering.  Some food processing clearly improved nutrient contents, compared to the 'fresh' alternative in shops (eg freezing of vegetables and fish and meat), and addition of micronutrients (sometimes for food preservation) might actually be valuable. The addition of iodine to milk, as a contaminant, has probably had a huge benefit for fetal development in countries with low environmental iodine such as UK.  

Bottled carbonated drinks may have been influential in preventing a great deal of gastroenteritis when fresh water is poor in quality, especially among children who may be unable to boil water.

This reviewer would value a clear account, perhaps a table of the evidence, and evidence-quality that links UPPs to chronic diseases.  As far as I am aware, the only evidence is observational, and all is heavily confounded by multiple socially-related factors with known effects on CD risk.  If that is the case, the authors need to acknowledge this weakness.  

The manuscript does not address environmental effects of UPPs.  That might be a more important and better evidenced aspect.

The tables are very difficult to read, with centred text and big gaps such that following horizontal rows is very exhausting.  Better layout is needed.

Author Response

Thank you for your comments and suggestions. Please find attached a response to your review. 

Reviewer 2 Report

This is an instructive paper, scientifically sound and assists in some way in the field of health-related research. It is clearly written despite the complexity of the subject matter.

I agree with the authors of the article, while there are numerous studies that examine the link between ultra-processed foods and drinks and human health, little is known, however, about the impact of the consumption of fortified ultra-processed foods on the overall diet composition and the risks of non-communicable diseases in children and adults. Therefore, I consider that the problem included in it is of great importance since the changes in the global food system production affect the whole world, and mainly low- and middle-income countries.

I assess positively the structure and content developed by the authors. However, in my opinion, more detailed information could be included on bibliographic search strategies, data sources and geographical information, and whether the inclusion and exclusion criteria have been considered.

Most of the literature referred to in the manuscript for the critical evaluation of ultra-processed products as food fortification alternatives comes from extensive scientific studies conducted in Latin American countries. In this situation, you need to think about and change the title of the manuscript and objective.

The list of suggested changes follows:

  1. Line 41: remove the comma between the end of the sentence and the bibliographic reference.
  2. Line 47: replace the comma with a full stop after the bibliographic reference.
  3. Lines 67, 89, 165, 173, 181, 187 209, 220 and 283: include a space between the bibliographic reference and just the previous word.
  4. Table 1: write the full stop at the end of the sentences in columns 2 and 3, some of them do not have a full stop.
  5. Lines 78 and 94: normalize the format of the bibliographic reference with or without space.
  6. Table 2: the superscript letter in the title of the last column does not match the explanation at the bottom of the table.
  7. Line 167: remove the space between the bibliographical reference and the full stop of the sentence.
  8. Line 177: remove the comma after the percentage symbol in the parenthesis.
  9. Between lines 202 and 203: “Panel 1. Fortification of ultra-processed product”

The information in the example included in Panel 1 in table format is confusing, has not been referenced in the text above and below and may be repetitive.

  1. Line 210: remove the space between the parenthesis and the comma. Add quote mark after calcium.
  2. Line 283: put a full stop at the end of the sentence.
  3. Lines 284-293 need a reference.
  4. Line 286: there is a 1 listed as a super index and there is no explanation.
  5. The format of the bibliography section should be standardised, specifically indication of page numbers. In most cases, only the page numbers are given, but in some cases pp (numbers: 2, 8, 12, 13, 19, 20, 23, 35, 43, 47, 61, 66, 72, 74, 75, 78, 79, 81.

Author Response

(The authors gave the same response as above.)

Round 2

Reviewer 1 Report

No new comments about the paper content.  Many people feel the term 'Ultra-Processed' is misleading and value laden.  It would be better to describe exactly what is meant, and not to lump together all the very different forms of food processing.